# OpenABM-Covid19—An agent-based model for non-pharmaceutical interventions against COVID-19 including contact tracing

Robert Hinch[1][*], William J. M. Probert[1], Anel Nurtay[1], Michelle Kendall[1,2], Chris Wymant[1], Matthew Hall[1], Katrina Lythgoe[1], Ana Bulas Cruz[1], Lele Zhao[1], Andrea Stewart[1], Luca Ferretti[1], Daniel Montero[3], James Warren[3], Nicole Mather[3], Matthew Abueg[4], Neo Wu[4], Olivier Legat[4], Katie Bentley[5,6], Thomas Mead[5,6], Kelvin Van-Vuuren[5], Dylan Feldner-Busztin[5], Tommaso Ristori[7], Anthony Finkelstein[8,9], David G. Bonsall[1,10], Lucie Abeler-Dörner[1], Christophe Fraser[1,11]

1 Big Data Institute, Li Ka Shing Centre for Health Information and Discovery, Nuffield Department of Medicine, University of Oxford, Oxford, United Kingdom, 2 Department of Statistics, University of Warwick, Warwick, United Kingdom, 3 IBM United Kingdom, Portsmouth, United Kingdom, 4 Google Research, Mountain View, California, United States of America, 5 The Francis Crick Institute, London, United Kingdom, 6 Department of Informatics, Kings College London, London, United Kingdom, 7 Department of Biomedical Engineering, Boston University, Boston, Massachusetts, United States of America, 8 Department of Computer Science, University College London, London, United Kingdom, 9 Alan Turing Institute, London, United Kingdom, 10 Oxford University NHS Trust, University of Oxford, Oxford, United Kingdom, 11 Wellcome Centre for Human Genetics, University of Oxford, Oxford, United Kingdom

☯ These authors contributed equally to this work.

* robert.hinch@bdi.ox.ac.uk

## Abstract

SARS-CoV-2 has spread across the world, causing high mortality and unprecedented restrictions on social and economic activity. Policymakers are assessing how best to navigate through the ongoing epidemic, with computational models being used to predict the spread of infection and assess the impact of public health measures. Here, we present OpenABM-Covid19: an agent-based simulation of the epidemic including detailed age-stratification and realistic social networks. By default the model is parameterised to UK demographics and calibrated to the UK epidemic, however, it can easily be re-parameterised for other countries. OpenABM-Covid19 can evaluate non-pharmaceutical interventions, including both manual and digital contact tracing, and vaccination programmes. It can simulate a population of 1 million people in seconds per day, allowing parameter sweeps and formal statistical model-based inference. The code is open-source and has been developed by teams both inside and outside academia, with an emphasis on formal testing, documentation, modularity and transparency. A key feature of OpenABM-Covid19 are its Python and R interfaces, which has allowed scientists and policymakers to simulate dynamic packages of interventions and help compare options to suppress the COVID-19 epidemic.

**Data Availability Statement:** All data generated by the model are available without restriction from the following repository - https://github.com/BDI-

pathogens/OpenABM-Covid19 - as the analysis described in this paper is fully reproducible. The data used to parametrise the model are publicly available, with all sources stated and linked to in the manuscript and its Supporting Information files. The observed hospitalisation data for Fig 5 are from UK govt coronavirus dashboard available here: https://coronavirus.data.gov.uk/ The seroprevalence data for Fig 5 are from ONS: https://www.ons.gov.uk/peoplepopulation andcommunity/healthandsocialcare/conditions anddiseases/articles/coronaviruscovid19 infectionsurveyantibodydatafortheuk/ 3february2021.

**Funding:** This work was funded by the Li Ka Shing Foundation (www.lksf.org), through an award to C. F. (funding the contributions of R.H, W.P., M.K., C. W., M.H., A.B.C., L.Z., A.S., L.F., D.B., L.A.D., and C.F.)and by research grant funding from the UK Department of Health and Social Care (DHSC), through an award to C.F. (funding the contributions of C.W. D.B. L.A.D., L.F., M.K., R.H. and C.F.). A.N. is funded by the ARTIC Network (Wellcome Trust Collaborators Award 206298/Z/17/Z). K.B., T.M., K. V.V. and D.F.B. were supported by the Francis Crick Institute which receives its core funding from Cancer Research UK (FC001751), the UK Medical Research Council (FC001751), and the Wellcome Trust (FC001751). For the purpose of Open Access, the author has applied a CC BY public copyright licence to any Author Accepted Manuscript version arising from this submission. The funders had no role in study design, data collection and analysis, decision to publish, or preparation of the manuscript.

**Competing interests:** I have read the journal's policy and the authors of this manuscript have the following competing interests: M.A., N.W., and O.L. are employees of Alphabet, Inc., a provider of the Exposure Notification System; no other relationships or activities that could appear to have influenced the submitted work. All other authors have declared that no competing interests exist.

## Author summary

Throughout the COVID-19 pandemic, computational modelling has been used to inform key uncertainties facing policymakers such as the number of cases and deaths, hospital capacity, tests and contact tracers. Models need to be: sufficiently complex to yield realistic predictions; computationally efficient to allow calibrations; and easy to use so that various policy mixes can be evaluated. OpenABM-Covid19 is a detailed epidemic model of the spread of COVID-19, simulating every individual in a population. Our model enables scientists and policymakers to quickly compare the effectiveness of non-pharmaceutical interventions like lockdowns, testing, quarantine, and digital and manual contact tracing. The model considers a hypothetical city with a default population of 1 million people whose ages and contact patterns are parameterised according to UK demographics. All of the parameters are openly documented and modifiable so that they can be adapted to fit other countries' data, and refined to match our understanding of COVID-19 as the epidemic progresses. The computer model simulates people's movement between their homes, workplaces, schools, and social interactions. OpenABM-Covid19 is open source and has been developed collaboratively by teams from academia and industry. Its modularity, documentation, testing framework, and accessibility via Python and R have provided validation, invited contributions, and encouraged wide adoption.

## Introduction

The novel coronavirus SARS-CoV-2 first appeared in China in late 2019 and spread across the globe in early 2020, causing several hundred thousand deaths world-wide in the first half of the year and overwhelming health systems [1]. Restrictions on movement were imposed in many countries, with severe impacts on social life, education, and economies [2]. Mathematical models have long been used to explain and forecast the course of epidemics and to predict the effects of public health interventions [3,4]. Most governments and policymakers use mathematical models to inform their decision-making [5]. The scientific community has responded by adapting old models and designing new models to learn more about the COVID-19 epidemic and inform public health.

Compared to compartmental models and branching-process models, agent-based models (ABMs) of the spread of infection allow for a more complete representation of the social contact network in which contagion occurs [6]. Major advantages include the ability to simulate heterogeneity in contact rates and local saturation effects, and the ability to better simulate contact tracing. Alongside other non-pharmaceutical interventions, contact tracing is an important intervention to help reduce the spread of COVID-19 [7,8]. In an ABM, the full history of all contacts can be stored, allowing for the impact of contact tracing to be explored in detail. For example, ABMs can include clustering in the contact network, so if incidence is high in a region of the contact network, an uninfected person who is contract-traced will be protected from this high level of local incidence. A downside of ABMs is that they are comparatively complex to code, are often not very parsimonious, and can be very computationally intensive to run, limiting the ability to explore a wide range of parameter combinations. ABMs have been used throughout the COVID19 pandemic to inform the public health response [9–14]. Here, we focus on developing OpenABM-Covid19, an agent-based simulation which addresses these downsides, by focussing on parsimony, computational efficiency, code transparency, and a robust testing framework.

A particular focus of our work applying OpenABM-Covid19 has been exploring different ways in which contact tracing, and in particular digital contact tracing using mobile phone

apps that record proximity events, can contribute to epidemic control [15]. Several other groups have approached this problem with similar ABMs [9,10].

We developed the agent-based model OpenABM-Covid19 to simulate an outbreak of COVID-19 in an urban environment. The default population is one million inhabitants with demographic structure based upon UK-wide census data, and household size and age-structure matched to data from the UK 2011 Census survey (for example, older people tend to live together and young children tend to live with younger adults).

On a daily basis all individuals in the model move between networks representing households and either workplaces, schools, or regular social environments for older people. Individuals also interact through random networks representing public transport, transient social gatherings etc. Membership of each type of network is determined by age, giving rise to age-assortative mixing patterns. Network parameters are chosen such that the average number of interactions match age-stratified data reported in [16]. The number of daily interactions in random networks is drawn from a negative binomial distribution, allowing for rare super-spreading events.

Infections are seeded in the population and spread through the networks. Biological and epidemiological characteristics of COVID-19 disease have been derived from the scientific literature. The model takes into account asymptomatic infections and different stages of severity, and includes the simulation of hospitalisations and ICU admissions. Since symptoms, disease progression and infectiousness are highly age-dependent, disease pathways in the model are age-stratified.

The ABM was developed to simulate different non-pharmaceutical interventions including lockdown, physical distancing, self-isolation on symptoms, testing and both manual and digital contact tracing. Modelling contact tracing requires the model to keep a record of previous interventions for a set number of days. A variety of contact tracing algorithms are included in the ABM, including tracing on symptoms and/or after a positive test, notifying first-degree contacts only or second-degree contacts as well, notifying household members or contacts of household members, testing of traced contacts, and imperfections in test-trace-isolate programmes such as delays, missed contacts and partial compliance. The model reports both aggregated data, such as incidence, tests required, individuals quarantined for various reasons, etc., and individual data such as transmission relationships.

OpenABM-Covid19 is available on Github (https://github.com/BDI-pathogens/OpenABM-Covid19), including model documentation, dictionaries for input parameters and output files, over 200 tests in a consistent testing framework used in model validation, and examples for running the model. The core of the model is implemented in the C language for speed; however, the model is run via Python using a SWIG-interface (see Implementation Details). This interface allows for dynamic intervention strategies to be modelled, as well as providing full transparency about the state of the model. This manuscript was prepared using v1.0 of the model and code for reproducing all figures in this manuscript from model output are publicly available online (https://github.com/BDI-pathogens/OpenABM-Covid19-model-paper).

OpenABM-Covid19 enables simulation of interventions to help policymakers determine the best options to suppress the COVID-19 epidemic in various settings. Default demographic parameters were chosen to reflect the UK and fit well to the COVID-19 epidemic in England; however, all parameters of the model can be changed by the user.

## Results

OpenABM-Covid19 was originally developed for evaluating the design of digital contact-tracing applications for the technology division of the National Health Service (NHSX) in April

2020 [15]. This work quantified the importance of rapid test processing and the level of user uptake required for digital contact-tracing to be effective. In a further study, OpenABM-Covid19 was used to investigate the benefits of deploying an application based on Google's Exposure Notification System (ENS) in Washington State [17]. This study included a calibration of the model to County-level data and demonstrated that digital contact-tracing provides benefits even when manual contact-tracing is deployed. OpenABM-Covid19 has been used throughout the pandemic by the National Health Service England (NHSE) to model hospital admissions at a regional level in England [18,19].

Whilst the principle aim of this paper is to give a detailed description of the model and its software interfaces, we now demonstrate some results of the model. First we look at some general features of the model including the interaction networks, the infection dynamics and the effect of population size. Next, we consider a case-study of the first wave of the COVID19 epidemic in England to demonstrate that a lightweight calibration can provide a close fit to several pieces of observed data. Finally, we simulate some of the intervention strategies which can be modelled by OpenABM-Covid19, including the requisite Python or R code, to demonstrate how complex multi-component intervention strategies can be simulated.

## General model properties

**Interaction networks.**   In each of the interaction networks, individuals are represented as nodes. Constant and dynamic connections occur between the nodes in the networks, representing interactions between individuals. The three networks represent different types of daily interactions: household, occupation (workplaces, schools or regular social environments for older people), and random (public transport, essential shopping, transient social gatherings) (Fig 1). The interaction networks have two roles in the ABM. First, the infection can be transmitted between two individuals on a day that they interact. Second, the interactions for each individual are stored and can be used for contact tracing. The membership of different networks leads to age-group assortativity in the interactions. Details of the construction of these three networks are given in the Methods. The distribution of the number of interactions on a simulated network by type and by age are shown in Fig 2A and 2B. Note how the mean number of contacts decreases with age as found in empirical studies [16]. The total number of interactions on the household network by age are shown in Fig 2C. Note how interactions are

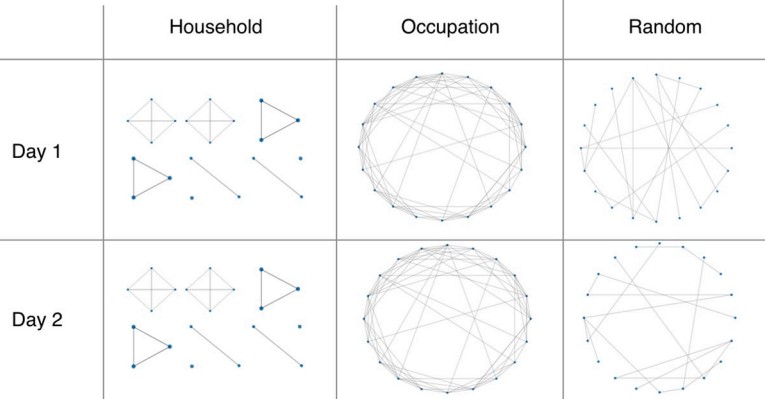

**Fig 1. Schematic depiction of the interaction networks within OpenABM-Covid19.** The household network is recurrent, the occupation network is a daily sample of a recurrent network, and the random network is transient and rebuilt each day.

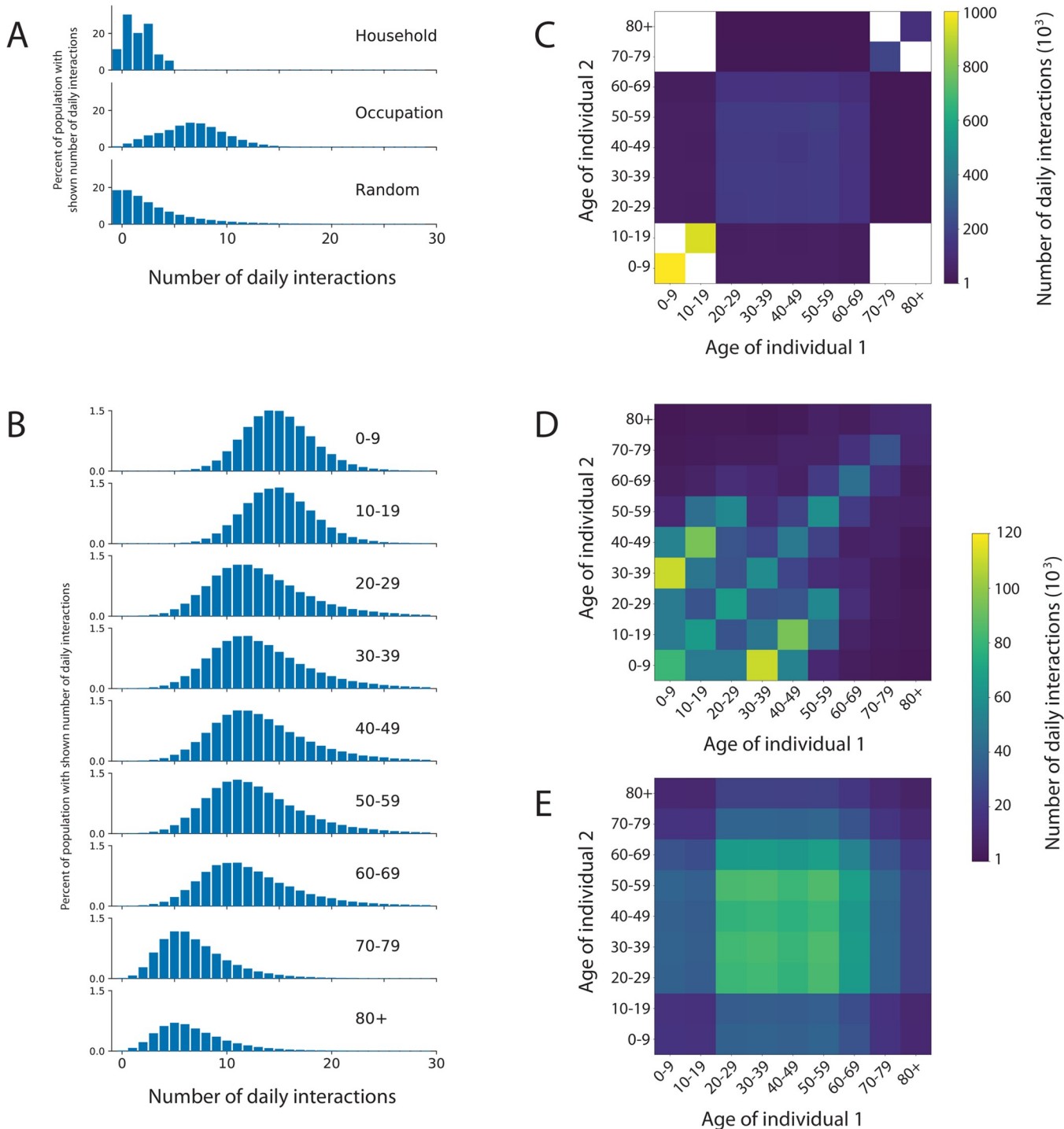

**Fig 2. Summary of interactions between individuals within OpenABM-Covid19.** (A) Distribution of daily simulated interactions stratified by the network upon which they occur. (B) Distribution of daily simulated interactions stratified by age group. Distribution of daily simulated interactions stratified by age group of both individuals in the (C) occupation, (D) household, and (E) random networks. Summarised interactions are from the first day of a single simulation in a population of 1 million individuals with UK-like demographics and household structure. Zero counts are shown in white in panels C, D, E.

clustered on the diagonal (people of the same age tend to live to live together) and the off-diagonal between children and adults aged 30 to 50 years (families).

**Infection dynamics.** The infection is spread by interactions between infected (source) and susceptible (recipient) individuals. The rate of transmission is determined by three factors: the infectiousness of the source, the age-dependent susceptibility of the recipient, and the type of interaction, i.e. on which network it occurred. Details of the infection model and how it was calibrated are in the Methods.

An example of how transmissions can be stratified by the infection status of the source and the age of both source and recipient is depicted in Fig 3. In this simulation of an uncontrolled epidemic, most transmissions occur from pre-symptomatic individuals with mild disease who are more numerous than individuals who go on to develop severe disease, followed by symptomatic individuals with mild disease. Interventions that reduce the rate of growth of

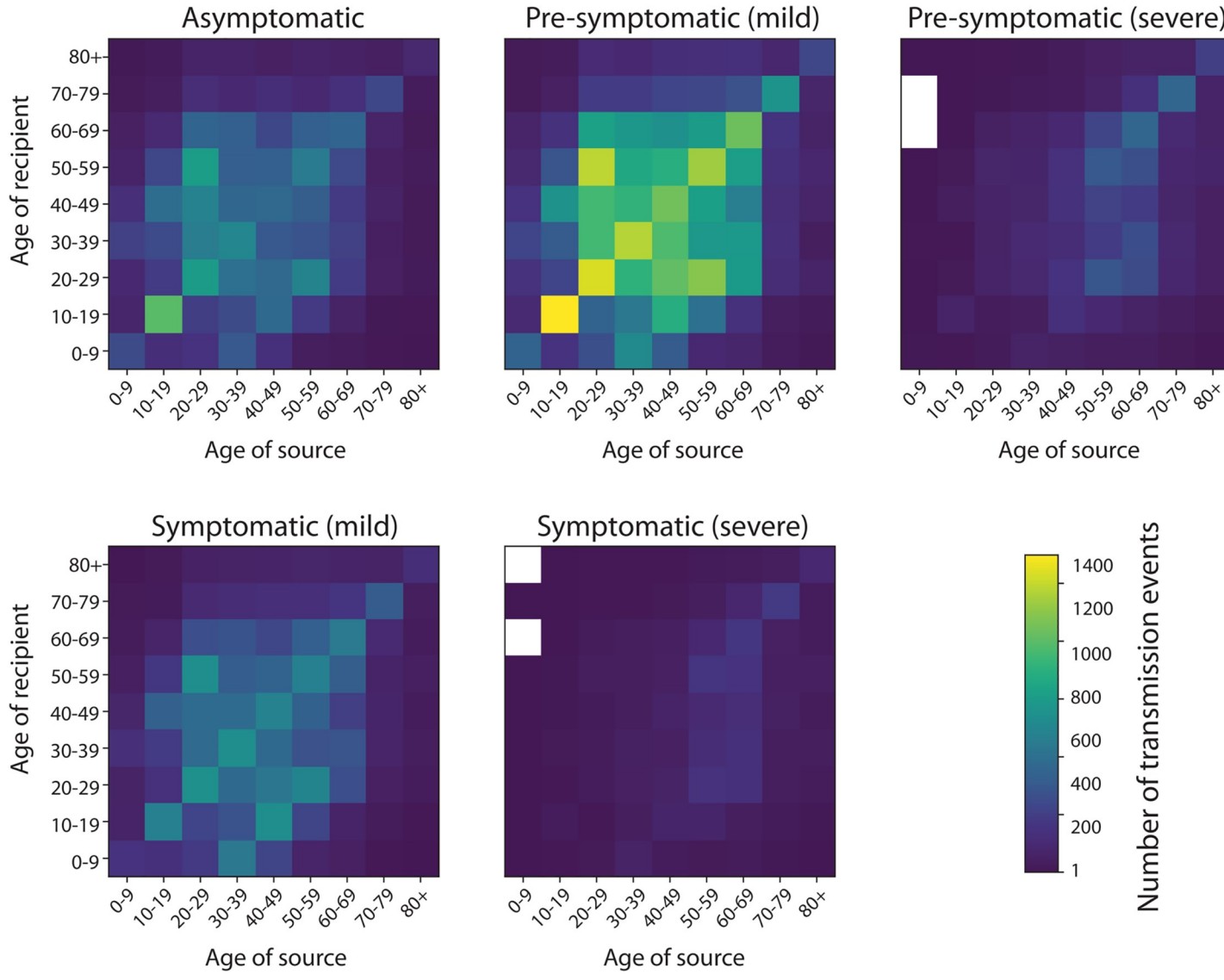

**Fig 3. Transmission events stratified by age of source and recipient and by infectious status of source.** Infectious status of source is specified in panel title. Data are from a single simulated epidemic of 1 million individuals with OpenABM-Covid19 following the first wave of the COVID19 epidemic in England. Zero counts are shown in white.

transmission will change the relative contributions of different symptomatic stages. Note that the largest number of transmissions occur pre-symptomatically before a mild infection in adults and children of secondary school age.

An important property of the epidemic is the offspring distribution which quantifies the amount of super-spreading. The offspring distribution was calculated by counting transmissions from the first 1% of people infected (i.e. the first 10,000 individuals) during the initial exponential growth phase of a simulation (S19 Fig). A negative-binomial distribution was fit using maximum likelihood estimation and gave an estimate of k = 0.5. This estimate is in-line with empirical estimates from surveys which give the estimates of 0.49–0.52 [20], 0.35–1.18 [21] and 0.32–0.49 [22]. The simulation to estimate k was run using the R interface for Open-ABM and the code is shown in S20 Fig.

The household secondary attack rate was calculated by counting the number of intra-household transmissions from the first person infected in each household and who was also in the first 1% of people infected in the initial exponential growth phase of a simulation. The household secondary attack rate for the default parameterisation of the model is 25%, which lies within the range of empirical estimates from Germany (21%) [23], the US (24%) [24] and the Netherlands (28%) [25]. Note that estimates from China are lower [26], however, the European/US studies are more applicable for a UK simulation.

**Population-size effects.**   We investigate the sensitivity of the model on the total population and the effect of aggregating sub-populations by estimating the systematic and stochastic variability of key statistics of the epidemic (S16 Fig, details in legend). The analysis showed that the stochastic variation in doubling rate (mean 3.5 days) was less than 0.2 days and the total number infected (mean 85%) less than 0.5% for simulations with at least 1 million people. The results showed no measurable difference in the mean value of the statistics for populations greater than 50k. Additionally, there was no measurable difference between running a simulation on a single population and aggregating across sub-populations with the same total population.

### Simulated epidemic for first wave in England (spring 2020)

The model was run on a population of 56 million people with UK demographics for the first wave of the COVID19 epidemic in England, by aggregating 56 simulations of 1 million people. An infection was seeded and grew exponentially with a doubling time of 3.5 days. A nationwide lockdown was introduced when prevalence reached 1.55% and the model then ran for a further 77 days. Several metrics are presented using the same parameterisation but for a simulation of 1 million individuals so that results can quickly be reproduced on a modern laptop or desktop computer. The age-dependent infection fatality ratio (IFR) for a representative simulation is depicted in Fig 4 and presented in Table 1, and is in line with other studies (e.g. [27,28]); other age-dependent outcomes are shown in S1 and S2 Figs. S3 Fig shows the corresponding waiting time distributions.

The main outputs of the model include the number of infected individuals, hospitalisations, ICU admissions and deaths, and for the aggregate simulations of 56 million individuals these can be compared to observed data for England (Fig 5). Additional outputs are the number of people in quarantine and the number of tests required, which is of particular interest when comparing different interventions. Transmissions can be depicted according to their type (pre-symptomatic, symptomatic and asymptomatic). The model provides a good fit to data on the first wave of the COVID19 epidemic in England with minimal calibration, matching the timing of the peak in daily deaths to within a few days, the trajectory of COVID19 patients in hospital beds, peak in hospital admissions, and national estimates of seroprevalence by early

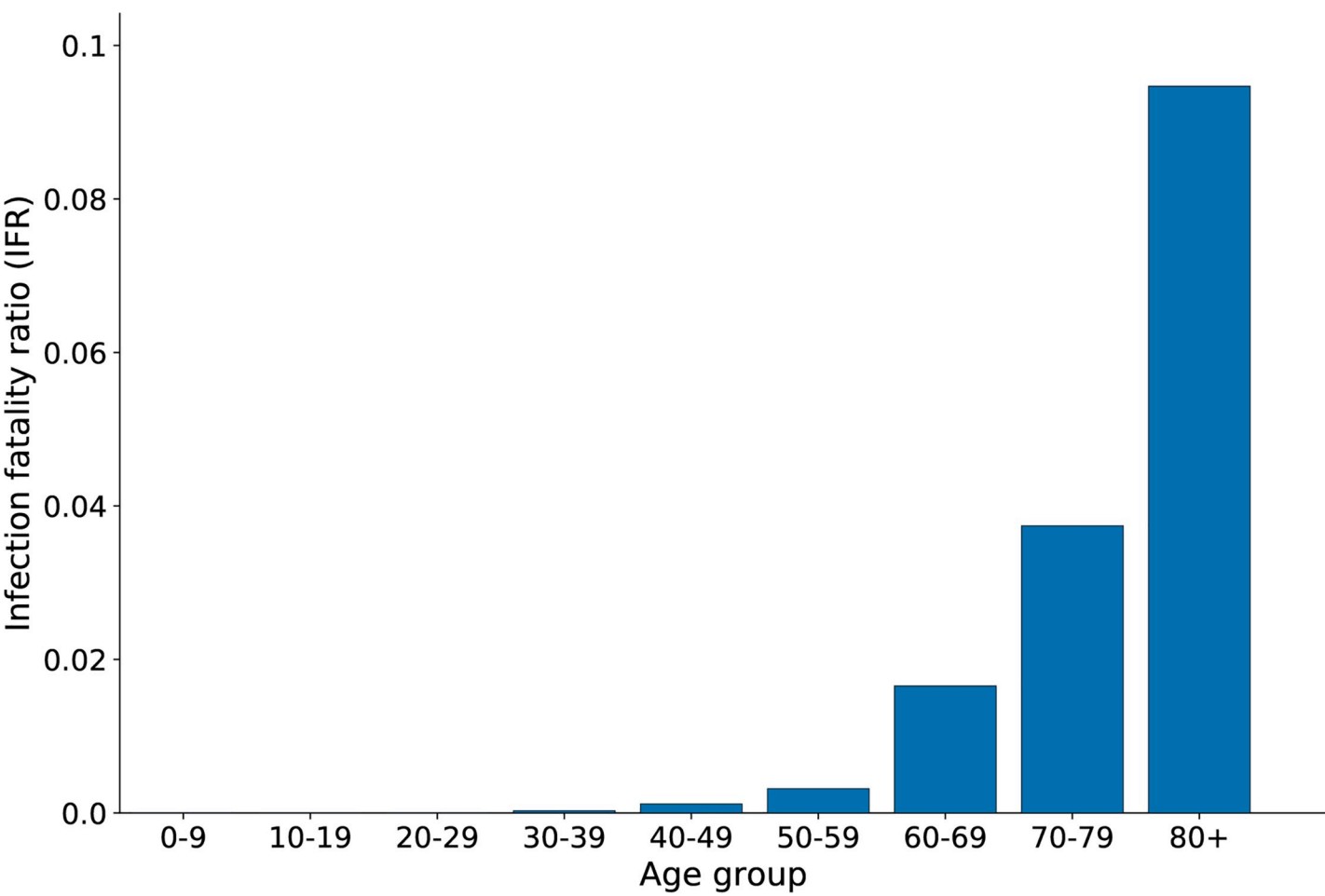

**Fig 4. Age-stratified infection fatality ratio (IFR) from a single simulation of OpenABM-Covid19.** Simulation in a population of 1 million with UK-like demography and with a lockdown when SARS-CoV2 prevalence reached 1.55%.

**Table 1. Age-stratified infection fatality ratio (IFR) from a single simulation of OpenABM-Covid19.**

| Age group | IFR (%) |
|---|---|
| 0–9 | 0 |
| 10–19 | 0 |
| 20–29 | 0 |
| 30–39 | 0.0292 |
| 40–49 | 0.1173 |
| 50–59 | 0.3165 |
| 60–69 | 1.655 |
| 70–79 | 3.7406 |
| 80+ | 9.4691 |
| Whole population | 0.8659 |

Simulation in a population of 1 million with UK-like demography and with a lockdown when SARS-CoV2 prevalence reached 1.55%.

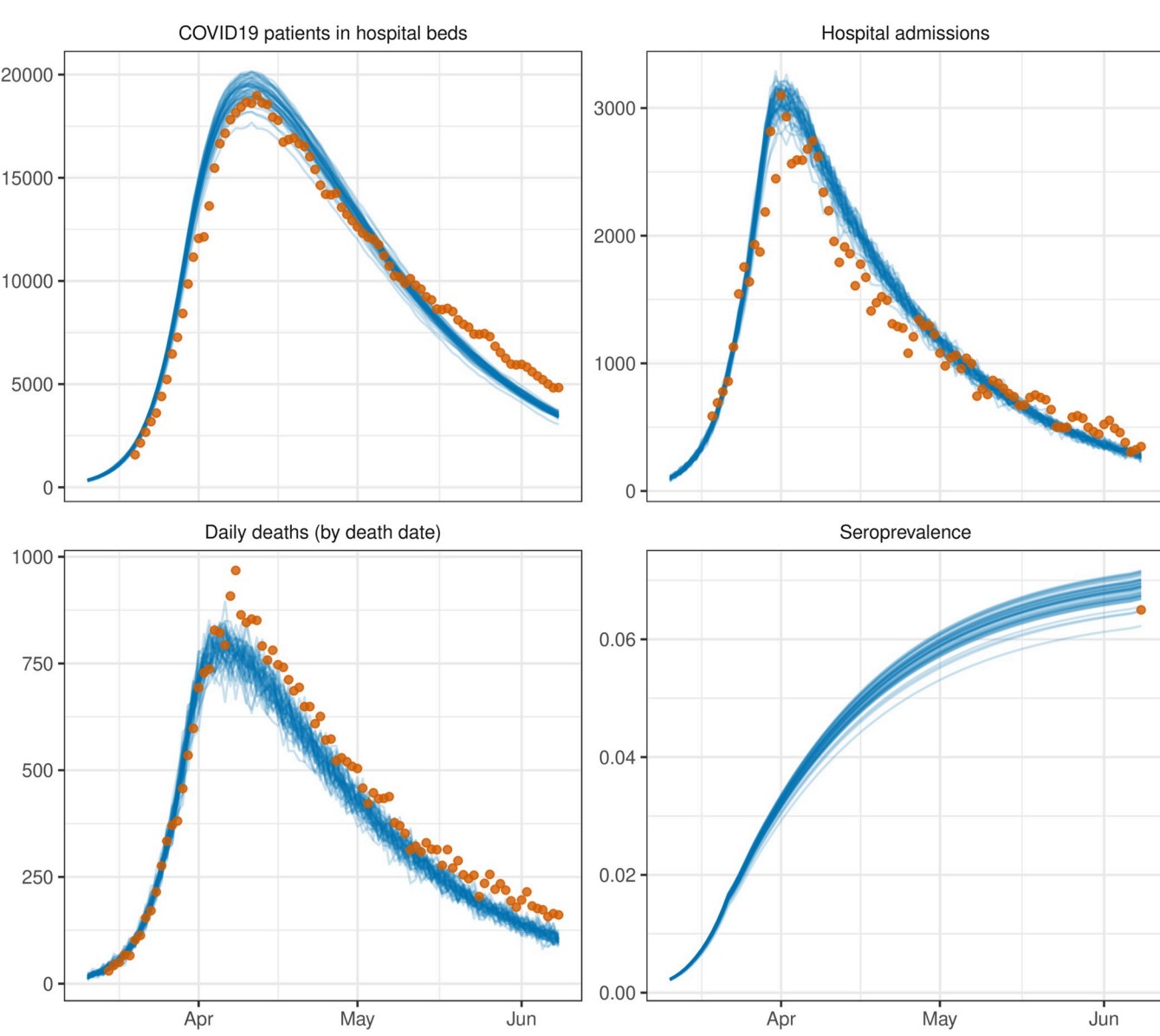

**Fig 5. Example of model outputs from OpenABM-Covid19 compared to observed data from the first wave in England.** Simulations are from 50 simulations in a population of 56 million individuals with UK-like demographics and control interventions. The beginning of the national lockdown is 23rd March 2020. Overlaid data are provisional counts of the number of deaths (measured by date of death) involving the coronavirus (COVID-19) registered in England (accessed on 5th June 2020), COVID19 patients in hospital beds (England), daily hospital admissions (England) from the UK government's COVID19 dashboard, and estimates of seroprevalence in England from the UK Office of National Statistics. Simulations are not calibrated to hospitalisation data, only shown for completeness.

June 2020 (Fig 5). Calibration involved fitting a transmission parameter (infectious_rate) so that doubling time of deaths of 3.5 days was matched (estimates of the doubling time in the UK were between 3–4 days [29]), and a two-dimensional grid search was then performed across the prevalence at which a national lockdown was implemented (calibrated to 1.55%) and the reduction in daily contacts under lockdown (calibrated to 0.33 of pre-lockdown levels; similar to values reported in [30] from the first wave in the UK).

## Non-pharmaceutical interventions and vaccinations

OpenABM-Covid19 can model a range of non-pharmaceutical interventions. Given the many types of intervention and interest in introducing them at different times, the interventions are controlled in the simulation dynamically through the Python interface. This allows for policy interventions to be applied in response to change in the growth of the epidemic (e.g. stricter policies such as lockdown can be applied when prevalence is above a threshold). Below we give brief descriptions of the interventions and sample Python code is given in the S5–S12 Figs with links to Jupyter Notebooks. All model parameters involved with non-pharmaceutical interventions are given in S10 and S11 Tables.

**Self-isolation upon symptoms.** A proportion of individuals self-isolate upon developing symptoms. Self-isolation is modelled by stopping interactions on the individual's occupation network and greatly reducing their number of interactions on the random network. The default time for self-isolation is 7 days with a daily dropout. The ABM contains the option to quarantine everybody within the household of the symptomatic individual. The ABM also considers individuals without COVID-19 who develop flu-like symptoms and may therefore self-isolate. S5 Fig is a Jupyter Notebook demonstrating how self-isolation upon symptoms reduces the rate of spread of the infection.

**Hospitalisation.** Once admitted to hospitals, a patient immediately stops interacting with the household, occupation and random networks. In the default model, we do not model interactions within hospitals. A preliminary module for hospital interactions has been developed and will be refined in future work.

**Lockdown.** Lockdown is modelled by reducing the number of interactions that people have on their occupation and random networks (by default by 67%). Additionally, given that during lockdown people stay at home, the transmission rate for interactions on the household network is increased by a factor of 1.5. S6 Fig is a Jupyter Notebook demonstrating the rapid reduction in the instantaneous reproduction number, R, when a lockdown is imposed. The impact of lockdown on the instantaneous and actual R is given in S13 Fig and an animation showing the age-stratified infection and disease compartments is in S1 Video.

**Shielding.** Contact reductions can be applied to certain age groups only. For example, given that fatality ratio is highly skewed towards the over 70s, we have the option of applying a reduction in contacts to this demographic group only. S7 Fig is a Jupyter Notebook demonstrating how new infections can be kept low in a shielded group.

**Physical distancing.** Measures such as physical distancing and mask wearing will reduce the probability of transmission in certain types of interactions (i.e. random interactions but not household interactions). The ABM allows for this to be modelled by allowing for the network-specific transmission multipliers to be adjusted during a simulation. S8 Fig is a Jupyter Notebook demonstrating how new infections can be kept low after a lockdown with (extreme) physical distancing measures.

**Testing and contact tracing.** OpenABM-Covid19 is able to model contact tracing (both manual and digital) and how it operates with or without an integrated testing system. The model contains many of the real-world imperfections which affect test and contact tracing programmes, such as test sensitivity and specificity, delays in testing and contact tracing, incomplete coverage, failure to recall contacts, contact tracer resource limitations and impartial adherence to quarantine requests. It also has the ability to model recursive contact tracing with and without testing. Below we give descriptions of the test and contact tracing features, with sample code given in S9–S12 Figs along with links to Jupyter Notebooks.

**Testing for SARS-CoV-2 infection.** Testing can occur in both the community and hospital (where an immediate clinical diagnosis is allowed). Tests are assumed to be sensitive from 3

days post-infection to 14 days post-infection with a default sensitivity of 80% and specificity of 99%. For community testing, delays can be introduced for ordering a test and for receiving the test result. Testing of an individual in the community is triggered by reporting symptoms and can also be triggered by being contact traced. S9 Fig demonstrates the importance of quick testing if self-isolation only occurs after a positive test (as opposed to on symptoms). The time-series output of the model shows both total infections (regardless of a test being performed) and total cases (those who have tested positive).

**Digital contact tracing.**    Contact tracing is vital to control epidemics with a high level of pre-symptomatic transmission. A variable fraction of individuals in each group can be assigned to have a digital contact tracing smartphone app. Ownership of smartphones is based on age-stratified OFCOM data (S4 Fig and S9 Table). Digital contact tracing can only occur between two app users. Digital proximity sensing is likely to miss some interactions, so when contact tracing a number of interactions are randomly dropped. For contact tracing, the model takes into account all interactions the individual has had with other app-users for the past seven days which have not been dropped. The model can simulate different app-based contact tracing algorithms. The app can send out notifications with the request to quarantine based on symptoms, or based on a positive test result of the index case. It can ask the household members of the index case and/or household members of the contacts to quarantine and also send out notifications deeper into the network if desired. It can request tests for contacts of index cases if desired. S10 Fig demonstrates how digital contact tracing following rapid testing can prevent a second wave even when the average uptake is at only 50% of the total population. S11 Fig demonstrates the calculation of the benefit to individuals of digital contact tracing.

**Manual contact tracing.**    Manual contact tracing works in a similar way to digital contact tracing with a few key differences. First, since it does not rely on an individual being a smartphone user, it can originate from anybody who tests positive (particularly important in the elderly where smartphone usage is lower). However, since the identification of interactions relies on the index case recalling them, only a fraction of actual interactions are traced. In particular, the fraction of interactions recalled depends on the type of interaction (i.e. occupation based interactions are more likely to be recalled than random interactions). Manual contact tracing only occurs after a delay following a positive test, to account for contact tracers contacting both the index and traced individuals. Finally, during a peak in the epidemic the amount of contact tracing required increases and risks overwhelming a manual contact tracing service. Therefore the model contains constraints on the total number of interviews that contact tracers can perform on a single day. S12 Fig demonstrates how a well-staffed manual contact tracing following rapid testing can lessen a second wave.

**Quarantine.**    Contact traced individuals can be asked to quarantine (default 14 days) either because they are directly traced or because they are a household member of somebody who has been traced. Like self-isolation, quarantine is modelled by stopping interactions on the workplace network and greatly reducing the number of interactions on the random network. The model includes a daily dropout rate to simulate imperfect adherence. Quarantine can be ended if the index case later tests negative (after tracing based upon their symptoms), or if the quarantined individual tests negative.

**Vaccination.**    OpenABM-Covid19 has the ability to model the effect of vaccination programmes in controlling the epidemic. Two types of vaccines are modelled: full protection where an individual cannot be infected; or protection from symptoms, where an individual can be infected but is asymptomatic. Efficacy is modelled as all-or-nothing for each individual, and for those who gain protection there is a delay between inoculation and the time at which the vaccine gives protection. The Python and R interfaces allow a vaccine schedule by age to be specified and multiple types of vaccine can be applied in a single simulation. S17 Fig

demonstrates the effect of a vaccine programme (both full and from-symptoms protection) where 2% of the adult population are vaccinated each and S18 Fig shows the R code used to simulate a bespoke vaccination programme.

## Discussion

We present OpenABM-Covid19, a COVID-19-specific agent-based model suitable for simulating the epidemic in different settings and assessing non-pharmaceutical interventions, including contact tracing using a mobile phone app. The model is well documented with a simple interface, allowing non-experts to easily evaluate complex dynamic intervention strategies in a few lines of Python or R code. OpenABM-Covid19 is an open-source project and is easily extensible, with new features already being added by multiple external teams. The model is fully documented and is thoroughly tested in a formal testing framework.

The model was designed to be as parsimonious as possible, with complexity only added when it was essential to model important features of COVID-19 or details of non-pharmaceutical interventions, and with parameters being inferred from published studies. Due to the substantial pre-symptomatic and asymptomatic transmission of the virus, it is necessary to model each individual's normal daily interactions. Further, on developing symptoms or during interventions such as contact tracing, the interaction pattern of individuals changes to only include those in the household. We therefore took the decision to model interactions using three social networks (household/occupational/random) with non-pharmaceutical interventions affecting each network differently. Recurring small-world networks were used to model interactions at home and at work, whereas a transient random network was used to model other daily interactions such as on public transport or in shops. The strong association of COVID-19 disease progression with age along with the age assortativity of social networks, led us to use a decade age-structure. The model simulated an urban population of 1 million rather than the population of a whole country to allow realistic estimates for hospitalisation and ICU admission forecasts on a regional level. Nevertheless, its use can be extended to perform an analysis on a country-wide level, as in Fig 5 for England. Large national epidemics will also exhibit meta-population dynamics rather than the spatially unstructured mixing modelled here.

One of the key aims of OpenABM-Covid19 was to model non-pharmaceutical interventions and, in particular, different forms of contact tracing. The model of digital contact tracing allows for questions such as the role of: testing delays, different quarantine requests, compliance rates, recursive testing, and app uptake to be investigated. The model of manual contact tracing allows for questions such as resource limitations, partial contact recall and interview delays to be investigated. The vaccination model allows for questions such as the order in which people are vaccinated to be investigated. Importantly, due to the simple Python and R interfaces it is possible for non-experts to simulate all these features and to investigate the effect of applying multiple intervention policies at different stages of the epidemic.

The current version of the model does not include nosocomial transmissions, transmission in care-home settings, non-hospital deaths, gender/sex of individuals, comorbidities, or any geographical structure apart from that implicit within the three modelled networks. All of these limitations are being currently addressed by collaborators and will become available on the Github repository in the near future. For example, a preliminary hospital model has been created to characterise the effect of SARS-CoV-2 transmission between patients, health-care workers and the wider community. The hospital model allocates patients to general and ITU wards according to symptoms; and then models the interactions between patients and health-care workers within wards and the hospital as a whole. The hospital model is currently available on the Github repository.

Another important area to develop will be the modelling of multiple-variants of SARS-CoV2. The current version of the model supports multiple-variants with different transmission rates, but assumes complete cross-immunity between variants. Whilst this is a reasonable model for the B.1.1.7 variant, it will be insufficient for modelling the B.1.351 or P.1 variants. These limitations are being addressed by collaborators and will become available on the Github repository in the near future.

OpenABM-Covid19 is a versatile tool to model the COVID-19 epidemic in different settings and simulate different non-pharmaceutical interventions including contact tracing. OpenABM-Covid19 is a modular tool that will help scientists and policymakers weigh decisions during this epidemic. Our vision is that, with the help of the world-wide modelling community, it will develop into a family of models for infectious diseases that are at risk of causing pandemics in the future, adding to the international toolkit for epidemic preparedness.

## Methods

### Demographics

Within the ABM, individuals are categorised into nine age groups by decade, from "0–9 year" to "80+ years". Decades were used instead of broader age groups because of the strong age-structure of the disease progression. By default, the demographics of the ABM are set to UK national data for 2018 from the Office of National Statistics (ONS). The proportion of individuals in each age group is the same as that specified by the population level statistics in S1 Table. Since we only consider simulating the epidemics up to a year, we do not consider changes in the population due to births, deaths due to other causes, and migration.

### Interaction network

A previous study of social contacts for infectious disease modelling, based on participants being asked to recall their interactions over the past day, has estimated the mean number of interactions that individuals have by age group [16]. We estimate mean interactions by age group by aggregating data (S2 Table). Fig 2A depicts the resulting distribution of contacts by network and Fig 2B by age.

Every individual is assigned to live in a single household. The household network is formed by all members of every household interacting with each other every day. The distribution of household sizes is the ONS estimate for the UK in 2018 (S1 Table). In addition to the household size, the mix of ages in households is important since multi-generational households provide a path by which the infection can be transmitted from young to old. To model this we used a reference panel of 10,000 households taken by down-sampling the UK-wide household composition data from the 2011 Census produced by the ONS. The overall household structure was generated by sampling from the reference household panel with replacement and using rejection-sampling to match the aggregate statistics for the age demographics and household size. The rejection-sampling method sequentially adds sampled households to the network if the deviation between the aggregate statistics and the target values (also from the ONS) was less than an acceptance threshold. The acceptance threshold is reduced as the sampled network grows and the final network is only accepted if the aggregate demographic and household size statistics are within tolerance of the targets (sum square error $< 10^{-5}$). The household members then makes up the population in the model. The number of daily interactions on a simulated network within each household by age is shown in Fig 2D.

Each individual is also a member of a recurring occupation network to model school, workplace or social networks. The occupation networks are modelled as small-world networks [31]. The network has a fixed set of connections between individuals, and each day a random subset

(50%) of these connections are chosen as the interactions between individuals. When constructing the occupation networks, the ABM ensures the absence of overlaps between the household interactions and the local interactions on the small-world network. For children, there are separate occupation networks for the 0–9 year age group (i.e. nursery/primary school) and the 10–19 year age group (i.e secondary school). On each of these networks we introduce a small number of adults (1 adult per 5 children) to represent teaching and other school staff. Similarly for the 70–79 year age group and the 80+ year age group we created separate networks representing daytime social activities among elderly people (again with 1 younger adult per 5 elderly people to represent some mixing between the age groups). All remaining adults (the vast majority) are part of the 20–69 network. Due to the difference in total number of daily interactions, each age group has a different number of interactions in their occupation network. Parameters and values corresponding to the occupation network are shown in S3 Table. The number of daily interactions on a simulated occupational network by age is shown in Fig 2C.

In addition to the recurring structured networks of households and occupations, we include random interactions. These are drawn randomly each day, independent of previous connections. The number of random connections an individual makes is the same each day (in the absence of interventions), drawn at the start of the simulation from an over-dispersed negative-binomial distribution. This variation in the number of interactions introduces some "super-spreaders" into the network who have many more interactions than average. The mean numbers of connections were chosen so that the total number of daily interactions matched that from a previous study of social interaction [16]. The number of random interactions was chosen to be lower in children in comparison to other age groups. Interactions in the random network are listed in S4 Table. The number of daily interactions on a simulated random network by age is shown in Fig 2E.

OpenABM also allows users to specify their own networks which can be added in addition (or instead of) to the default networks in the model. An example of how to add a user specified network via the R interface is given in S13 Fig.

## Infection dynamics

Infectiousness varies over the natural course of an infection, i.e. as a function of the amount of time the source has been infected, $\tau$. Infectiousness starts at zero at the point of infection ($\tau = 0$), increases to a peak at an intermediate time, and decreases to zero a long time after infection (large $\tau$). Following [7], we took the functional form of infectiousness to be a scaled gamma distribution. We chose the mean and standard deviation as intermediate values between different studies [7,32,33]. Once infected, we split individuals into three groups based upon the eventual severity of the disease: asymptomatic, mild symptomatic and moderate-severe symptomatics. The level of infectiousness depends upon this grouping, i.e. pre-symptomatic individuals who go on to develop moderate-severe symptoms are more infectious than those who go on to develop mild symptoms. By default, the overall infectiousness of asymptomatic individuals and individuals with mild symptoms, is 0.33 and 0.72 times that of individuals with moderate-severe symptoms respectively [34].

The susceptibility of the recipient to infection is modelled with a scale factor dependent on the recipient's age. To determine these factors, we identified studies of whether or not transmission occurred from index cases to monitored close contacts [21,35–41]. Lower probability of infection in children was reported in almost all studies, including that of Zhang et al [35] which observed more infections than the rest of the studies combined, with consistent adjustment for other covariates of transmission risk. We used the susceptibility by age of Zhang

et al., interpolated to match our ten-year age categories. The merged data and fit are shown in S5 Table.

## Transmission dynamics

We model the type of interaction, i.e. on which of the three networks the interaction took place. Whilst we do not have data on the length of interactions, interactions which take place within a person's home are likely to be closer than other types of interactions leading to higher rates of transmission. This is modelled using a scale factor, which is 2 by default and gives a good estimate of the household secondary attack rate (see Results section). Finally, to fully account for the over-dispersion in offspring infections, we add an individual infectious factor which is drawn independently for each individual (and is the same for all their interactions). Combining all effects, we model the hazard rate per interaction at which the virus is transmitted by

$$\lambda(t, d, a, n) = \frac{LS_a A_d B_n G}{\bar{I}} \int_{t-1}^{t} f_\Gamma(u;\ \mu_i, \sigma_i^2) du$$

where $t$ is the time since the source was infected; $d$ indicates the disease severity of the source (asymptomatic, mild, moderate/severe); $a$ is the age of the recipient; $n$ is the type of network where the interaction occurred; $\bar{I}$ is the mean number of daily interactions; $f_\Gamma(u; \mu, \sigma^2)$ is the probability density function of a gamma distribution; $\mu_i$ and $\sigma_i$ are the mean and width of the infectiousness curve; $L$ scales the overall infection rate; $S_a$ is the relative susceptibility of the recipient based on age; $A_d$ is the relative infectiousness of the source based on disease severity; $B_n$ is the scale factor for the network on which the interaction occurred; and $G$ is the individual infectious factor which is drawn for each individual from a Gamma distribution with mean 1 and s.d. $\sigma_{II}$. S6 Table contains the values of the parameters used in simulations. The transmission hazard rate $\lambda$ is converted to a probability of transmission via $P = 1 - e^{-\lambda}$. The epidemic is seeded by randomly infecting individuals on the day before the simulation starts.

## Natural history of infection

Upon infection, an individual enters a disease progression cascade where the outcome and rates of progression depend on the age of the infected person. The disease state transitions are shown in Fig 6 and the model parameters in S7 and S8 Tables.

A fraction $\Phi_{asym}$(age) of individuals are asymptomatic and do not develop symptoms, a fraction $\Phi_{mild}$(age) will eventually develop mild symptoms, and the remainder develop moderate/severe symptoms. Each of these proportions depend on the age of the infected individual (S7 Table). Those who are asymptomatic are infectious at a lower level (see Infection Dynamics section) and will move to a recovered state after a time $\tau_{a,rec}$ drawn from a gamma distribution.

Once an individual is recovered the model allows immunity to wane through time using two parameters: a fixed period for which every individual must wait, $\tau_{waning-shift}$, and then a geometric distribution of waiting times until individuals become susceptible, parameterised by its mean $\tau_{waning-mean}$. By default, the model assumes $\tau_{waning-shift}$ to be 10,000 days (essentially no waning immunity). During this waiting period, infection is assumed to be completely immunising (recovered individuals cannot be reinfected).

Individuals who will develop symptoms start by being in a pre-symptomatic state, in which they are infectious but have no symptoms. The pre-symptomatic state is important for modelling interventions because individuals in this state do not realise they are infectious and

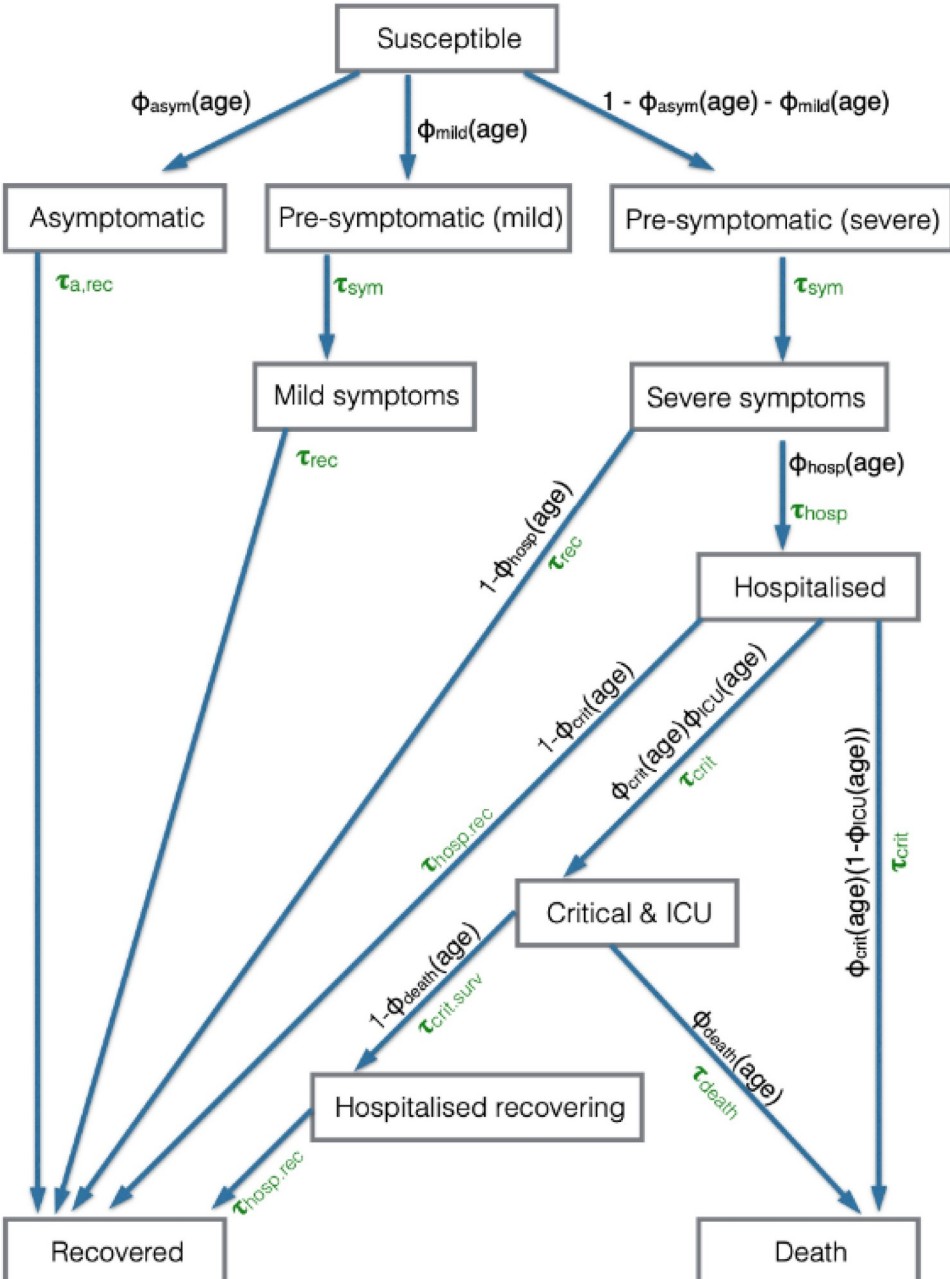

**Fig 6. Schematic of infection and disease transitions within OpenABM-Covid19.** The disease status of an individual, and the probability and time distribution of transitions. The $\Phi_{xxx}$(age) variables are the probability of transition to a particular state, when the individual can progress to more than one state within the model, where the probability depends upon the age of the individual. The $\tau_{xxx}$ are the gamma distributed variables of the time taken to make the transition.

therefore will not self-isolate based on symptoms to prevent infecting others. Individuals who develop mild symptoms do so after time $\tau_{sym}$ and then recover after time $\tau_{rec}$ (both drawn from gamma distributions). The remaining individuals develop moderate/severe symptoms after a time $\tau_{sym}$ drawn from the gamma distribution.

Whilst most individuals recover without requiring hospitalisation, a fraction $\Phi_{hosp}(age)$ of those with moderate/severe symptoms will require hospitalisation. This fraction is age-dependent. Those who do not require hospitalisation recover after a time $\tau_{rec}$ drawn from a gamma distribution, whilst those who require hospitalisation are admitted to hospital after a time $\tau_{hosp}$, which is drawn from a shifted Bernoulli distribution. Among all hospitalised individuals, a fraction $\Phi_{crit}(age)$ develop critical symptoms and require intensive care treatment, with the remainder recovering after a time $\tau_{hosp,rec}$ drawn from a gamma distribution. The time from hospitalisation to developing critical symptoms, $\tau_{crit}$, is drawn from a shifted Bernoulli distribution. Of those who develop critical symptoms, a fraction $\Phi_{ICU}(age)$ will receive intensive care treatment. For patients receiving intensive care treatment, a fraction $\Phi_{death}(age)$ die after a time $\tau_{death}$ drawn from a gamma distribution, with the remainder leaving intensive care after a time $\tau_{crit,surv}$. Patients who require critical care and do not receive intensive care treatment are assumed to die upon developing critical symptoms. Patients who survive critical symptoms remain in hospital for $\tau_{hosp,rec}$ before recovering.

OpenABM-Covid19 also includes a meta-population model which runs a simulation on parallel sub-populations (which can be parameterised separately). After each time step, new cases can be seeded in each region based upon the incidence of infections in connected sub-populations.

## Implementation details

The core of OpenABM-Covid19 is coded in C with object-oriented interfaces in Python and R. The code is written in a modular manner to ease readability and encourage extension of the code base. It is open source and is being actively developed by multiple teams. The model uses the GNU Scientific Library (GSL) for mathematical functions, statistical distributions, and random number generation [42] and so any distribution or function available within the GSL can be easily incorporated into the model (for instance in modelling waiting-time distributions). Memory is pre-allocated at the start of the simulation for efficiency.

An important feature of the implementation are the Python and R interfaces using SWIG, which is a package for providing interfaces between high-level languages and C/C++ [43]. Running the model via Python allows for complex dynamic interventions strategies to be easily modelled (see examples in S5–S12 Figs). All states of the model (e.g. transmission events, interactions, individual characteristics) are exposed in Python, which gives full transparency to the results of the model. For example, S12 Fig is a Notebook showing how to calculate the relative personal protective effect of app users versus non-app users when digital contact tracing is used. Python is also a ubiquitous language amongst data scientists, and the interface allows them to fully interact with the model whilst keeping the high speed and memory performance of C.

The metapopulation model is implemented in Python using the standard module *multiprocessing* [44], with each sub-population running in a separate process. Both the initial model set-up and the daily time-step are run in parallel, allowing for approximate linear speed-ups when running on a multi-core machine.

The model codebase includes over 200 tests used to validate the model. Each test ensures an expected output from the model is realised for a specified set of input parameters. Tests are written in a consistent manner, using the pytest framework. All tests are automatically run when new contributions to the codebase are made. Tests vary input parameters ensuring that expected behaviour of the model is realised across a wide range of input parameter values. Tests cover a range of domains including: disease dynamics, infection and transmission dynamics, non-pharmaceutical interventions, network construction, the C and Python

interface, the waiting time distributions, file concordance across the multiple output files from the model, and non-disease related demographics.

**Performance.**  The computational efficiency was measured by simulating an epidemic with default parameters until the total prevalence reached 1%; then reducing the transmission rates on the occupational and random networks so that R was reduced to 1; and simulating for 100 days in total. On a 2019 MacBookPro (2.4 GHz Quad-Core Intel Core i5), the simulation took 1.5s per day for a population of 1m people and scaled linearly with population size (S15A Fig). The time is dominated by daily rebuilding of networks, and running the simulation with static-networks took 50ms per day for a population of 1m people. The meta-population model of OpenABM with multiple sub-populations of 100k running in parallel threads took 250ms per day for a total population of 1m people. The meta-population model with static-networks took 15ms per day for a population of 1m people. The persistent memory used in the simulation was measured using the iprofiler command and Instruments. For a simulation where only 1 day of interactions are stored, the persistent memory is about 1kb per person (S15B Fig) and scales linearly with the population. The memory usage is split in approximate thirds between: networks; interactions; and data about individuals (e.g. timings of events such as infection). For a simulation where 7 days of interactions are stored (i.e. for contact-tracing modelling), the persistent memory is about 3kb per person of which about 80% is storing the interactions.

## Supporting information

**S1 Fig. Proportion of each age group ever infected, ever hospitalised, or deceased.** Simulations are from the end of a single simulation in a population of 1 million individuals with UK-like demographics and control interventions. The simulation was run for 77 days after lockdown started. The denominator in each calculation is the number of individuals in each age group in the total population (e.g. of the total population, the middle panel shows the proportion of each age group that was hospitalised in the simulation).
(TIF)

**S2 Fig. Age-stratified hospital admissions, ICU admissions, and deaths.** Data are from a single simulation of 1 million individuals with UK-like demographics and control interventions. The simulation was run for 77 days after lockdown started. The denominator is the number of individuals ever having been in the state in question (e.g. of all simulated hospitalisations, the top panel shows the distribution of these by age).
(TIF)

**S3 Fig. Waiting time distributions for transitions between infection and disease states.** All distributions are gamma except time to hospital which is a shifted Bernoulli distribution. Mean of each gamma distribution is shown with a vertical dashed line.
(TIF)

**S4 Fig. Smartphone usage by age in the UK.**
(TIF)

**S5 Fig. Example notebook of self-isolation on symptoms.** Code for this notebook is provided at https://github.com/BDI-pathogens/OpenABM-Covid19-model-paper/blob/master/notebooks/example_self_isolation.ipynb.
(TIFF)

**S6 Fig. Example notebook of a lockdown.** The Python code used for this simulation, the code is also at https://github.com/BDI-pathogens/OpenABM-Covid19-model-paper/blob/master/

notebooks/example_lockdown.ipynb.
(TIFF)

**S7 Fig. Example of lockdown followed by shielding.** Code for this notebook is provided at https://github.com/BDI-pathogens/OpenABM-Covid19-model-paper/blob/master/notebooks/example_lockdown_shield.ipynb.
(TIFF)

**S8 Fig. Example lockdown followed by social distancing.** Code for this notebook is provided at https://github.com/BDI-pathogens/OpenABM-Covid19-model-paper/blob/master/notebooks/example_lockdown_social_distance.ipynb.
(TIFF)

**S9 Fig. Example of self-isolation after testing.** Code for this notebook is provided at https://github.com/BDI-pathogens/OpenABM-Covid19-model-paper/blob/master/notebooks/example_testing.ipynb.
(TIFF)

**S10 Fig. Example of digital contact tracing.** Code for this notebooks is provided at https://github.com/BDI-pathogens/OpenABM-Covid19-model-paper/blob/master/notebooks/example_digital_contact_tracing.ipynb.
(TIFF)

**S11 Fig. Example app user protection calculation.** Code for this notebook is provided at https://github.com/BDI-pathogens/OpenABM-Covid19-model-paper/blob/master/notebooks/example_digital_contact_tracing_protect.ipynb.
(TIFF)

**S12 Fig. Example of manual contact tracing.** Code for this notebook is provided at https://github.com/BDI-pathogens/OpenABM-Covid19-model-paper/blob/master/notebooks/example_manual_contact_tracing.ipynb.
(TIFF)

**S13 Fig. Reproduction number.** Data from a single simulated outbreak with R calculated using the complete simulated transmission tree (actual) or using the time series (instantaneous). Simulation data are for a single simulation in a population of 1 million individuals with UK-like demographics. The vertical dashed mark where interventions were introduced (self-isolation on symptoms followed by lockdown), note that $R_{actual}$ is reduced prior to the introduction of each intervention.
(TIF)

**S14 Fig. Example adding a user defined network.** R script demonstrating how to add a user specified network. The R code is at: https://github.com/BDI-pathogens/OpenABM-Covid19-model-paper/blob/master/R/figS14_example_add_network.R.
(TIFF)

**S15 Fig. Performance.** A. The computation time per day for different size populations. The default networks are dynamic and are rebuilt each day, whereas the static networks are not changed after the first day. The meta-population model was run on a quad-core processor. B. The required memory for a simulation which is linear in population.
(TIFF)

**S16 Fig. Scaling and stochastic variation.** Simulations for epidemics were run for different population sizes and split into different numbers of equal sub-populations in meta-models

(zero case migration). Initially the epidemic was seeded with 0.05% infections and an uncontrolled epidemic was allowed to develop for 100 days, with approximately no new infections at the end. A minimum of 20k people was required in each subpopulation in order for there to be sufficient seed infections to prevent a stochastic extinction at the start. Each simulated epidemic was characterised by 2 basic statistics: the doubling time in days to go from 1% to 2% of the population infected; and the total fraction of the population infected. Each configuration was run 10 times and the figure is a box plot of the results, with the number of subpopulations shown as separate colours. The simulations show that the mean doubling time and fraction infected are roughly independent of the total population size. The stochastic variation is determined by the total population and is independent of the number of subpopulations. With a total population of at least 1 million people, the stochastic variation in the doubling time was <0.2 days and in the total number infected was <0.5%.
(TIFF)

**S17 Fig. Vaccine result.** A simulation of a vaccine programme implemented after a lockdown period to control the epidemic. The epidemic was allowed to grow until 2% of the population had been infected, at which point a lockdown was implemented for 30 days along with a vaccination programme where 2% of adults were inoculated each day (vaccine 90% effective after 15-days). The figure compares the total deaths and infections for a vaccine which offers full protection from symptoms, to one which only offers protection from symptoms and to no vaccine programme. The R code for generating this figure is at https://github.com/BDI-pathogens/OpenABM-Covid19-model-paper/blob/master/R/figS17_vaccine.R.
(TIFF)

**S18 Fig. Vaccine R code.** R used for generating the simulations with vaccination programmes. The R code is at https://github.com/BDI-pathogens/OpenABM-Covid19-model-paper/blob/master/R/figS18_example_vaccination.R.
(TIFF)

**S19 Fig. Offspring distribution.** The offspring distribution (in blue) and the sibling distribution (in grey). A negative-binomial fitted to the offspring distribution gives the estimate of $k = 0.51$. The inset is the cumulative sibling distribution against the cumulative offspring distribution and shows that the 70% of infections are generated by the top 20% of individuals.
(TIFF)

**S20 Fig. Offspring distribution R code.** R script used for generating the offspring distribution is at https://github.com/BDI-pathogens/OpenABM-Covid19-model-paper/blob/master/R/figS20-offspring-distribution.R.
(TIFF)

**S21 Fig. Mean square error comparing simulated and observed data during the first wave of the COVID-19 epidemic in England across four data sources.** Simulations are of 56 million individuals, performed separately for a grid of values across a two-dimensional grid of 1) prevalence of SARS-CoV-2 at which lockdown was implemented (y axis), and 2) reduction in daily contacts during lockdown (x-axis). Surface has been interpolated from a grid of values. Transmission parameters (infectious_rate) fixed to assume a doubling time of approximately 3.5 days. Red dots highlight those parameter sets with the smallest 5% error with observed data. Observed data are from the UK Governments COVID19 dashboard and the UK's Office of National Statistics (seroprevalence).
(TIF)

**S22 Fig. Epidemic doubling time (in daily deaths) as a function of infectious_rate parameter.** Simulations of 56 million individuals using OpenABM-Covid19 across a range of values of the infectious_rate parameter (black dots) across a range from 3.5 to 8.5 in increments of 0.1. Each black dot is the slope of fitting a linear regression to log of daily simulated deaths (truncated to between the first 100 to 1000 deaths). The red line represents a fit of the form a(x-b)^c * exp(nu), where nu is a noise term, to these data. Each red dot gives the value of the parameter infectious_rate (in brackets) for a doubling time of 3 (7.1), 3.5 (5.8), and 4 (5.0) days respectively.
(TIF)

**S1 Table. UK population stratified by age and UK households stratified by household size.** Data provided by the ONS. Parameter values match the OpenABM-Covid19 baseline parameters, April 25, 2021.
(CSV)

**S2 Table. Average number of non-household interactions stratified by age.** Values shown are for random and occupational interactions for an individual in each age group per day from empirical estimates [16]. Parameter values match the OpenABM-Covid19 baseline parameters, April 25, 2020.
(CSV)

**S3 Table. Occupational network parameters.** Mean numbers of daily occupational connections for members of each age group, fraction of adults involved in occupational networks for children and for elderly people, and rewiring parameters for randomisation of daily interactions. Parameter values match the OpenABM-Covid19 baseline parameters, April 25, 2020.
(CSV)

**S4 Table. Parameters for numbers of random connections that members of each age group have per day.** Parameter values match the OpenABM-Covid19 baseline parameters, April 25, 2020.
(CSV)

**S5 Table. Susceptibility by age.**
(CSV)

**S6 Table. Infection parameters.** The mean of the generation time distribution and the standard deviation of the infectious period were calculated from [7,45–48]. Parameter values match the OpenABM-Covid19 baseline parameters, April 25, 2020.
(CSV)

**S7 Table. Age-stratified infection and disease parameters.** Proportion of people in each stage of illness whose disease progresses further [Calibration of [49] & Spanish Serology Survey for fraction of asymptomatic and mild symptoms; Calibration of [27,49] & Spanish Serology Survey for fraction hospitalised; [27] for fraction of hospitalised that require critical care; [27,50–52] for fatality fraction]; [53,54]. Parameter values match the OpenABM-Covid19 baseline parameters, April 25, 2020.
(CSV)

**S8 Table. Parameters for waiting time distributions.** Mean and standard deviation for density functions of the times that each transition–disease progression or recovery–takes [29,51,55]. For the shape of the functions see S2 Fig. Parameter values match the OpenABM-Covid19 baseline parameters, April 25, 2020. * Personal communication with SPI-M; data

soon to be published.
(CSV)

**S9 Table. Smartphone usage stratified by age in the UK.** Data based on the Technology Tracker (fieldwork 9 Jan– 7 Mar 2020) and the Children's Media Literacy tracker (fieldwork 25 April– 11 July 2019), data provided by the Office of Communications.
(CSV)

**S10 Table. Parameters for hospitalisation and self-quarantine upon symptoms.** Parameter values match the OpenABM-Covid19 baseline parameters, April 25, 2021.
(CSV)

**S11 Table. Parameters corresponding to testing and contact tracing.** Parameter values match the OpenABM-Covid19 baseline parameters, April 25, 2020.
(CSV)

**S1 Video. Animation of a simulated outbreak.** Data from a simulated outbreak in a population of 1 million individuals with UK-like demographics and control interventions showing age-stratified histograms of individuals in each compartment. (https://github.com/BDI-pathogens/OpenABM-Covid19-model-paper/blob/d01baf1ca160aec649ce52c26001d8721dfb6bf9/figures/fig_outbreak_animation.mp4) Arrangement of sub-panels are in the same arrangement as the model compartments (Fig 4).
(MP4)

## Author Contributions

**Conceptualization:** Robert Hinch, William J. M. Probert, Anel Nurtay, Michelle Kendall, Chris Wymant, Matthew Hall, Katrina Lythgoe, Luca Ferretti, Matthew Abueg, Neo Wu, Katie Bentley, Thomas Mead, Kelvin Van-Vuuren, Dylan Feldner-Busztin, Tommaso Ristori, Anthony Finkelstein, David G. Bonsall, Lucie Abeler-Dörner, Christophe Fraser.

**Data curation:** Robert Hinch, William J. M. Probert, Anel Nurtay, Michelle Kendall, Chris Wymant, Ana Bulas Cruz, Lele Zhao, Luca Ferretti, Lucie Abeler-Dörner, Christophe Fraser.

**Formal analysis:** Robert Hinch, William J. M. Probert, Anel Nurtay, Ana Bulas Cruz, Lucie Abeler-Dörner, Christophe Fraser.

**Funding acquisition:** Lucie Abeler-Dörner, Christophe Fraser.

**Investigation:** Robert Hinch, William J. M. Probert, Anel Nurtay, Michelle Kendall, Chris Wymant, Matthew Hall, Katrina Lythgoe, Ana Bulas Cruz, Lele Zhao, Luca Ferretti, Anthony Finkelstein, David G. Bonsall, Lucie Abeler-Dörner, Christophe Fraser.

**Methodology:** Robert Hinch, William J. M. Probert, Anel Nurtay, Michelle Kendall, Chris Wymant, Matthew Hall, Katrina Lythgoe, Lele Zhao, Luca Ferretti, Daniel Montero, James Warren, Nicole Mather, Matthew Abueg, Neo Wu, Olivier Legat, Katie Bentley, Thomas Mead, Kelvin Van-Vuuren, Dylan Feldner-Busztin, Tommaso Ristori, Anthony Finkelstein, David G. Bonsall, Lucie Abeler-Dörner, Christophe Fraser.

**Project administration:** Robert Hinch, William J. M. Probert, Anel Nurtay, Ana Bulas Cruz, Andrea Stewart, Lucie Abeler-Dörner, Christophe Fraser.

**Resources:** Andrea Stewart, David G. Bonsall, Lucie Abeler-Dörner, Christophe Fraser.

**Software:** Robert Hinch, William J. M. Probert, Anel Nurtay, Michelle Kendall, Chris Wymant, Daniel Montero, James Warren, Nicole Mather, Matthew Abueg, Neo Wu, Olivier Legat, Katie Bentley, Thomas Mead, Kelvin Van-Vuuren, Dylan Feldner-Busztin, Tommaso Ristori.

**Supervision:** Robert Hinch, William J. M. Probert, Christophe Fraser.

**Validation:** Robert Hinch, William J. M. Probert, Anel Nurtay, Michelle Kendall, Chris Wymant, Matthew Hall, Ana Bulas Cruz, Daniel Montero, James Warren, Nicole Mather, Matthew Abueg, Neo Wu, Olivier Legat, Katie Bentley, Thomas Mead, Kelvin Van-Vuuren, Dylan Feldner-Busztin, Tommaso Ristori, David G. Bonsall, Christophe Fraser.

**Visualization:** Robert Hinch, William J. M. Probert, Anel Nurtay, Michelle Kendall, Matthew Hall, Ana Bulas Cruz, Lucie Abeler-Dörner, Christophe Fraser.

**Writing – original draft:** Robert Hinch, William J. M. Probert, Anel Nurtay, Michelle Kendall, Chris Wymant, Matthew Hall, Katrina Lythgoe, Lele Zhao, Luca Ferretti, David G. Bonsall, Lucie Abeler-Dörner, Christophe Fraser.

**Writing – review & editing:** Robert Hinch, William J. M. Probert, Christophe Fraser.

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
