## [Decision Letter · Decision Letter 0]

25 Jan 2021

Dear Bulas Cruz,

Thank you very much for submitting your manuscript "OpenABM-Covid19 - an agent-based model for non-pharmaceutical interventions against COVID-19 including contact tracing" for consideration at PLOS Computational Biology.

As with all papers reviewed by the journal, your manuscript was reviewed by members of the editorial board and by several independent reviewers. In light of the reviews (below this email), we would like to invite the resubmission of a significantly-revised version that takes into account the reviewers' comments.

We cannot make any decision about publication until we have seen the revised manuscript and your response to the reviewers' comments. Your revised manuscript is also likely to be sent to reviewers for further evaluation.

Sincerely,

Benjamin Muir Althouse

Associate Editor

PLOS Computational Biology

Virginia Pitzer

Deputy Editor-in-Chief

PLOS Computational Biology

Reviewer's Responses to Questions

**Comments to the Authors:**

Reviewer #1: The manuscrit describe a new computing agent-based code to simulate pandemic evolution, and to study some non-pharmaceutical interventions impact on this pandemic evolution. This approach allow to include some complex phenomena, such as heterogeneous social interactions, impact of physical distancing on the contaminations, contact tracing, social interaction in schools, at home, public transport, etc. This open-source code is writtend in C with a Python interface to allow various interactions between the user and the code, and for results display.

The authors are invited to add in the manuscrit some clarifications of the limitation of their model (~ line 391), and some precisions, as described bellow.

- the authors use full randomized contact network to simulate public transports (line 157). What is the impact of such approach on "real" contacts, where people have some habits ? Is a partially-randomized network feasible ?

- are the different interactions temporaly splitted to avoid overlap ? If no did you study the impact of the order of the different daily interaction computing on the evolution of the pandemy ?

- line 206 : 65 millions people simuled by aggregating 65 simulation of 1 million people. Are 65 non-interactive simulations similar than a simulation of 65 M agent where everybody can interact with everybody ?

- is an homogeneous spatialized distribution of people representative of real demographic distribution, where some regions are more dense than other ? (in particular for public transport and number of daily social interactions)

- what are the data used for model calibration ? Number of death ?

- line 270 : the lockdown is taken into account with a reduction of socials interactions of 71%. Is this value deduced to fit to the mesures or taken from a previous study ?

- line 283 : did you deduce the probability of transmission with and without physical distancing ?

- line 431 : every individual interact with each other every day. Same queestion than before : is this temporally splitted to avoid overlap contamination (A <-> B <-> C contribute each day of the contamination A <-> C) ?

- line 504 : is the 2 factor used is deduced to fit to the observed data or is this documented ?

- line 600 : is the code parallelized (OpenMP or MPI) ? Did he scales correctly (in term of computing time / number of agents, and in term of number of core used if paralelized) ?

- line 600 : 1.7 Gb memory for 1 million agents means ~ 400 simple precision (or ~ 200 double-precision) values used per agent without contact tracing. What are the main memory consuption to need so much RAM ?

- due to high stochastic characteristics of a pandemy evolution (due to the random contacts between agents, and the random contagion from an agent to another), did you study the reproductibility of the simulations, and the range of results obtained with exactly the same parameters ? Some others agent-based approach, which use fixed contact-network, can show a wide variation of the max number of infected people, only due to the random aspect of contagion at each social contact. A study on these variations needs to be added to the manuscrit (These effects are visible for initial conditions with low number of infected people). The results used for decisional help are they an average of many simulations, or only one-shot simulation, without any error bars due to the stochastics effects ? This clarification is imperative, and may need to balance the conclusions obtained by the simulation described in the manuscrit.

- what are the initial conditions ? Some infected at t = 0 ? How many ?

Reviewer #2: The paper describes the implementation of an ABM for simulating the spread of Covid-19 within a population of up to 1 million people, although as it is described this number could be increased. Spreading occurs through both static and dynamic contact networks using statistics on contact patterns, which addresses the inhomogeneous spread in different age-groups. Also, disease progression and asymptomatic cases are modelled. The strength of the presented simulation model is the inclusion of a range of different prevention measures in addition to the classic NPIs for contact reduction. As this is a core advantage of this implementation, systematic assessment and testing of intervention strategies should be performed and discussed more deeply.

• The authors state that the model is parameterized for the UK, however, the paper lacks information on the calibration and validation process and how well the model reproduces the historic epidemic curve and efficiency of actual NPIs.

• The model seems to be able to reproduce the characteristics of a certain (?) time interval of the COVID epidemic in the UK. Line 229 "The model provides a good fit to UK data, correctly matching: the cumulative number of deaths; the magnitude of the peak in daily deaths; the timing of the peak in daily deaths to within a few days; and peak hospitalisation to within 25% of the recorded number" This could be supported with additional quantification of errors/correspondence in the comparison between real data and simulation results. How was this correspondence achieved? How good is the quality of fit compared with other models? Several times in the text the authors mention that the model (and parameters) were calibrated, but a description or discussion of the calibration approaches is not found in the manuscript.

• The line "The model was run on a population of 65 million people with UK demographics, by aggregating 65 simulations of 1 million people. An infection was seeded and grew exponentially with a doubling time of 3.5 days." It should be described why does the model fits nevertheless? Based on the methods, the doubling time is a result of the simulation. This should be explained and why is it (?) constant?

• Paragraphs with important "messages" should be revised thoroughly. To give a more or less random example in line 600 ff "Performance. The ABM for 1 million individuals takes approximately 3s per day to run…" It is described in the abstract, but missing at this point for what scenario and which time range. Such shortcomings can be resolved easily and can improve the quality of the paper to a large extent.

• Evaluation of results of prevention measures is provided in supporting information only (see comments below). The analysis and assessment of interventions should be systematic with visual displays supporting a thorough discussion. In the current state, intervention scenarios are somehow treated as technical demonstrations.

• The abstract description of the model is well written and insightful, but technical details and modelling decisions are not or provided. The appendix is very useful and the parameters are well described.

• Even though established approaches in an agent-based simulation of epidemic spread and the state of the art (and its limitations as a motivation for the paper) should be recognized (if it is the goal of the paper to show the methodological improvement). The motivation for modelling decisions and assumptions should be discussed in the presentation of a simulation model. E.g. motivate the use of certain types of networks/graphs and probability distributions! What are the features of the specific mathematical concepts? Why are they suitable to model certain aspects?

• The paper lacks technical details on implementation for assessing whether the efficiency claims hold, however as it is open source it is possible to review the code itself.

• A technical overview of the implementation as a simulator/framework should be provided. E.g. what is an "object-oriented programming style"? To the knowledge of the reviewer, C is not an object-oriented programming language. What is the procedure for sampling a population? A description of the initialization phase would be helpful. How are individuals aggregated into households on a technical level? How are the networks sampled from data? This should be included despite the source code is publicly available and well documented.

• In the discussion, a kind of outlook is included, which should be described in more detail. A focus is set on hospitals, but big importance on epidemiological modeling will be set on the possibility to include also pharmaceutical interventions like vaccination

• How is the model dealing with "unreported cases"? As it is immanently clear, this aspect should be at least mentioned.

The simulation model is of high quality and shows also high potential. But at the current state, the paper is a conceptual presentation of a simulation model but does not include or present interesting results on one of the three areas 1) epidemiology, 2) HCI and Usability, or 3) technical novelty. The paper should increase focus towards one of those directions. If the model can be adapted as easily as claimed by the authors the model could provide a good framework for additional research for the assessment of NPIs if the model is correctly parameterized for the addressed research question.

Some of the figures in the manuscript look not very appealing. For instance, Fig1 could be supported by context if placed in the Methods section. The inclusion of parameters is important. All mathematical symbols should also be used in the parameter tables. It would be good to provide additional context to the parameter values with some of the supporting tables.

The model is fully documented and is thoroughly tested. The formal testing framework (mentioned in line 358) could be described in more detail e.g. line 589 "The model codebase includes over 200 tests used to validate the model." What was included in the tests? How was validation implemented?

Model descriptions and modelling process could be set concerning international standards or guidelines like "Modelling Good Research Practices of ISPOR-SMDM" (https://www.sciencedirect.com/science/article/pii/S109830151201652X) . For example "V-8 If using an agent-based model, thoroughly describe the rules governing the agents, the input parameter values, initial conditions, and all sub-models."

Reviewer #3: This paper presents a detailed description of the OpenABM-Covid19 model, which is an agent-based COVID-19 transmission model informed by detailed data on contact networks and validated against data from the UK. As a methods paper, it does not include results per se, but rather illustrates the analyses the model can be used for.

Overall, I found the paper to be exceptionally well written and clearly laid out. The model has been carefully conceived, and the use of modern software practices (testing, documentation, concern for adaptability, an easy-to-use Python user interface, etc.) make this study an exemplar for how such models should be developed and communicated. The following comments are mostly intended as nonbinding suggestions for improving the paper.

p. 4, line 56: This could be interpreted to mean that the population size fixed at 1 million, rather than that 1 million is the default.

p. 5, line 85: I'm not sure I understand how people who are contact-traced are themselves "protected" -- wouldn't they be contact traced following exposure to a known positive?

p. 6, line 96: Quite a few groups have developed COVID ABMs; while a comprehensive literature review is probably beyond the scope of the introduction, additional citations of influential ABMs might help the reader better understand the modeling landscape. The following are suggestions only:

* The Imperial model, which was influential in UK policy decisions (https://www.imperial.ac.uk/media/imperial-college/medicine/sph/ide/gida-fellowships/Imperial-College-COVID19-NPI-modelling-16-03-2020.pdf)

* Blakeley et al., which was influential in Australian policy decisions (https://www.mja.com.au/journal/2020/213/8/probability-6-week-lockdown-victoria-commencing-9-july-2020-achieving)

* Koo et al. (https://www.thelancet.com/journals/laninf/article/PIIS1473-3099(20)30162-6/fulltext)

* Aleta et al. (https://www.nature.com/articles/s41562-020-0931-9)

* Rockett et al. (https://www.nature.com/articles/s41591-020-1000-7)

In addition, the specific claims about OpenABM-Covid19 in comparison to other models may not be entirely accurate. Rockett et al. and Bicher et al. used 23 million and 9 million agents, respectively, which are larger populations than are typically used in OpenABM-Covid19. In addition, my understanding is that Covasim's computational efficiency is comparable to OpenABM-Covid19 (1 second per 2 million person-days; see Fig. S6 of https://www.medrxiv.org/content/10.1101/2020.07.15.20154765v4.full.pdf), and has also been designed with extensibility and ease of development in mind (e.g. 100 forks on GitHub). The authors are encouraged to rephrase in a way that emphasizes the strengths of OpenABM-Covid19 while also acknowledging the strengths (and weaknesses!) of other models. (Disclosure: I am one of the authors of Covasim.)

p. 7, line 111: While I can see a (strong) argument for community and perhaps workplace contacts to be drawn from a negative binomial distribution, is this true of household and school contacts as well? We have found that overdispersion in infectiousness, rather than overdispersion in number of contacts, is the most important factor for driving superspreading events. If in your model the latter are alone sufficient to account for the observed distribution of secondary infections, this is an interesting finding!

p. 8, line 139: Out of curiosity, is there any reason why this version cannot be considered 1.0? The codebase seems mature, tested, and documented, and the bulk of development seems to have been completed >6 months ago, which would seem to exceed the threshold for a 1.0 release in most contexts. (Very minor point: the Python package installs as version 0.2, not 0.3.) While acknowledging that the software practices are light-years ahead of most models, I did find myself wishing for a changelog, at least for backwards-incompatible changes (i.e. if the same parameters would no longer run, or would give a different result).

p. 10, Figure 2:

1. For Fig. 2A, showing the three distributions separately might be easier to read (as is, it looks like some of them are negative).

2. I am also surprised at how low the number "random" contacts are -- I assume this does not count the 50+ people one would be in "contact" with on the metro or at a grocery store.

3. I didn't realize until getting to the Methods section that school students were included in the "occupation" network. This surprised me since school class sizes and workplace sizes tend to have fairly different distributions (the former having a larger mean and smaller variance than the latter). In addition, given the central importance of school closures as a COVID policy measure, including school networks explicitly would seem to be desirable. Unless this can be added quickly, it could be noted as a limitation of the model.

p. 11, line 208: Typo, "day"

p. 11, Fig. 4: Perhaps a few words could be said about how the model-derived IFR compares to empirical estimates, e.g. Ferguson et al. (which seems like it was used for some of the input parameters), O'Driscoll et al. (https://www.nature.com/articles/s41586-020-2918-0), and/or Brazeau et al. (https://www.imperial.ac.uk/mrc-global-infectious-disease-analysis/covid-19/report-34-ifr). By eye at least, the match looks quite good (i.e., at least as consistent as these estimates are with each other), which is nice evidence for model validation.

p. 13, Fig. 5:

1. What process, if any, was used for calibrating the parameters to produce these outputs? It seems some relatively straightforward tuning would be able to produce a better fit (e.g., lowering the infection hospitalization rate; increasing the hospitalization mortality rate). But if no tuning was done at all to parameters, the fact that the parameter "priors" produce such a good fit is worth highlighting.

2. X-axis labels are missing -- I assume this is days? If actual date labels could be used, it would make it easier to read.

3. It would be interesting to know how well the model can fit the UK's 2nd and 3rd waves, but one can easily get lost in an infinite spiral of fitting the model to the latest data, so I would not consider doing this a requirement.

4. Does the model produce estimates of diagnoses, and if so, would it be possible to see the projections for these as well?

5. Some indication of uncertainty would be valuable -- stochastic uncertainty if not parametric uncertainty. I understand that this will be highly dependent on the number of seed infections used, however.

6. The commit hash mentioned here does not match the previous one, even though this figure seems to be the central result of the paper. Does the previously mentioned commit hash refer to the IFR results? Would one get different results if running with a different commit hash (e.g., 536adae at the time of writing)?

p. 18, line 365: If I'm not mistaken, it has recently become possible to run from R as well as Python? (Although this seems less well documented.) This might be worth mentioning given the large number of epidemiologists who use R.

p. 20, line 394: It is also probably worth mentioning pharmaceutical interventions, since I see vaccination has also been recently implemented in the model.

p. 20, line 403: I was surprised there was no mention in this paper of the contexts and applications OpenABM-Covid19 has been used for. I feel silly saying this, but citing https://www.medrxiv.org/content/10.1101/2020.08.29.20184135v1 would seem to be needed at minimum!

p. 23, line 468: It sounds like some individuals are assigned a high daily number of contacts and persist with that number of contacts. It may be more realistic to redraw the number of contacts per person each day as well since superspreading events tend to happen at non-daily venues (e.g. churches, restaurants). However, this is unlikely to make much difference to the results. In addition, it would be interesting to see the distribution of primary vs. secondary cases, such how well the model matches the data that e.g. ~20% of people are responsible for ~80% of transmissions.

p. 28, line 571: I am curious to know why it's claimed that the approach is object-oriented -- C is not generally considered an object-oriented programming language (https://softwareengineering.stackexchange.com/questions/113533/why-is-c-not-considered-an-object-oriented-language). I see structs being used extensively to handle data of different types, but I don't see much use of pointers to functions being used to emulate class methods, for example, and as such it looks a bit more functional to me. (Of course, one could "just" recode the whole thing in C++!)

p. 28, line 579: It might help to explain what SWIG is (I had to google it).

p. 29, line 603: Is it possible to disable rebuilding the daily interaction network? I imagine this would increase performance by perhaps an order of magnitude, and can approximated as a larger network with lower transmission probabilities (i.e., 10 new contacts per day for 5 days each with a 1% transmission probability is virtually identical to 50 static contacts for 5 days with an 0.2% transmission probability -- given the wide uncertainty bounds of how many contacts should exist in the first place).

p. 52, line 814: Perhaps consider a protected branch instead of a commit hash, e.g. https://github.com/BDI-pathogens/OpenABM-Covid19-model-paper/blob/ploscb/notebooks/example_digital_contact_tracing.ipynb

-- Cliff Kerr

**Have all data underlying the figures and results presented in the manuscript been provided?**

Reviewer #1: Yes

Reviewer #2: Yes

Reviewer #3: Yes

PLOS authors have the option to publish the peer review history of their article (what does this mean?). If published, this will include your full peer review and any attached files.

Reviewer #1: No

Reviewer #2: No

Reviewer #3: **Yes: **Cliff Kerr
---

## [Decision Letter · Decision Letter 1]

4 Jun 2021

Dear Bulas Cruz,

We are pleased to inform you that your manuscript 'OpenABM-Covid19 - an agent-based model for non-pharmaceutical interventions against COVID-19 including contact tracing' has been provisionally accepted for publication in PLOS Computational Biology.

Best regards,

Benjamin Muir Althouse

Associate Editor

PLOS Computational Biology

Virginia Pitzer

Deputy Editor-in-Chief

PLOS Computational Biology

Reviewer's Responses to Questions

**Comments to the Authors:**

Reviewer #1: The new version answer all my questions.

Reviewer #2: Thank you very much for dealing with our comments. I think all important aspects have been adressed and changes have been implemented.

Reviewer #3: The authors have done an excellent job responding to all points. I was especially pleased to see the changes related to the code, including the performance section and the vaccination interventions.

**Have the authors made all data and (if applicable) computational code underlying the findings in their manuscript fully available?**

Reviewer #1: Yes

Reviewer #2: Yes

Reviewer #3: Yes

PLOS authors have the option to publish the peer review history of their article (what does this mean?). If published, this will include your full peer review and any attached files.

Reviewer #1: No

Reviewer #2: No

Reviewer #3: **Yes: **Cliff Kerr

---

## [Editor Report · Acceptance letter]

8 Jul 2021

PCOMPBIOL-D-20-02225R1 

OpenABM-Covid19 - an agent-based model for non-pharmaceutical interventions against COVID-19 including contact tracing

Dear Dr Bulas Cruz,

I am pleased to inform you that your manuscript has been formally accepted for publication in PLOS Computational Biology. Your manuscript is now with our production department and you will be notified of the publication date in due course.

With kind regards,

Katalin Szabo
